# Understanding the Role of Cancer Diagnosis in the Associations between Personality and Life Satisfaction

**DOI:** 10.3390/healthcare11162359

**Published:** 2023-08-21

**Authors:** Weixi Kang, Edward Whelan, Antonio Malvaso

**Affiliations:** 1UK DRI Care Research and Technology Centre, Department of Brain Sciences, Imperial College London, London W12 0BZ, UK; 2Independent Researcher, 99MX QH Maynooth, Ireland; 3Department of Brain and Behavioral Sciences, University of Pavia, 27100 Pavia, Italy

**Keywords:** cancer, personality, Big Five, life satisfaction, moderation

## Abstract

Life satisfaction refers to the degree a person enjoys their life. An integrated account of life satisfaction is discussed in the literature, which proposes that life satisfaction is made up of personality traits and areas of life satisfaction (e.g., satisfaction with health, job, and social life). In addition, disruptions in one domain (e.g., health) may disrupt the association between personality traits and life satisfaction. The current research was interested in if clinically diagnosed cancer could influence the association between the Big Five personality traits and life satisfaction. The current study analyzed data from 1214 people with a diagnosis of cancer (38.55% males) with an average age of 59.70 (S.D. = 15.53) years and 13,319 people without a cancer diagnosis (38.13% males) with an average age of 59.97 (S.D. = 11.10) years who participated in Understanding Society: the UK Household Longitudinal Study (UKHLS). For the first time, our study revealed that cancer markedly influences the relationship between Agreeableness and life satisfaction, after accounting for demographic variables. Neuroticism was negatively associated with life satisfaction in people with and without clinically diagnosed cancer, whereas Agreeableness and Extraversion were positively associated with life satisfaction in people with and without clinically diagnosed cancer. Openness and Conscientiousness were positively related to life satisfaction in people without cancer but were not significant predictors in people with cancer. Health professionals should develop strategies and interventions by fostering personality traits, including Agreeableness, Openness, Conscientiousness, and Extraversion, while reducing Neuroticism.

## 1. Introduction

Life satisfaction is a construct that describes how people enjoy their lives [1]. Concepts such as well-being, happiness, and life satisfaction are synonymous in the literature [2]. While this may not be correct, there are undoubted similarities between these concepts [3]. Happiness focuses on positive emotions and immediate experiences of joy and contentment, which is difficult to objectively measure [3]. Some assert that life satisfaction and the related concept of well-being are easy to measure [4]. Subjective well-being is a comprehensive and multifaceted concept that encompasses various aspects of an individual’s life. Amongst others, Diener’s model proposes that subjective well-being has two parts: cognitive and affective well-being [5]. Affective well-being includes happy and unhappy mental states, while life satisfaction belongs to the cognitive aspect of subjective assessments of well-being [6].

There is a need to understand the factors that influence life satisfaction, in particular for those with chronic conditions including cancer. This can support the design of interventions that lead to enhanced reports of life satisfaction that contribute to a superior quality of life, which may have many benefits such as reducing the risk of mortality [7] and disability [8,9] and promoting social interactions [10,11].

Theories of life satisfaction mainly include the top-down, bottom-down, and integrated models [1,2]. The top-down approach suggests that life satisfaction is a result of an individual’s innate personality characteristics. This is produced by positive cognitive and affective states that help with people’s ability to cope and function [12]. While the bottom-down approach constructs life satisfaction as being determined by variables such as positive perceptions of various aspects of one’s life such as one’s health, economic status, social capital, sociability, and housing [1,2,12,13]. Finally, an integrated account of life satisfaction suggests that life satisfaction is made up of the interplay of personality and various aspects of life satisfaction [1]. Regarding the relationships between personality traits captured by the Big Five model, Neuroticism can negatively impact life satisfaction. Personality characteristics such as Agreeableness, Openness, Conscientiousness, and Extraversion are seen as helping with life satisfaction (e.g., [1,2,14]), although not always [15,16,17,18,19,20,21,22].

Recent research on the topic shows that an integrated model of life satisfaction is most helpful in understanding life satisfaction [1,2]. This approach holds that positive reports of life satisfaction are a result of factors, such as aspects of life satisfaction, demographic traits, and personality characteristics. Life satisfaction is stable over a lifetime [23,24,25], and this is seen as supporting the top-down approach, which believes that personality traits are critical. However, life satisfaction can be irrevocably altered because of life crises such as bereavement or unemployment [26,27]. One disappointing experience can lead to a re-evaluation of one’s level of life satisfaction, which can lead a person reassessing their life. For example, a diagnosis of a chronic illness such as cancer can prompt individuals to reassess their careers [1,2]. This could be said to show that the relationship between personality characteristics and life satisfaction, as argued in the integrated theory, is never fixed but is dynamic [1,2].

The primary demographic factors that contribute to an individual’s level of life satisfaction include age, gender, income, education, and marital status. Various studies consistently show that women, married individuals, and those with higher incomes tend to report higher levels of life satisfaction [28,29,30,31,32,33,34]. However, there are conflicting findings regarding the relationships between age, education, and life satisfaction. Some studies find no significant association between age and life satisfaction [35], while others report either a positive [36] or negative [37] correlation. Notably, Helliwell and Putnam [38] demonstrated that individuals aged 65 and above tend to have higher life satisfaction compared to younger individuals. Similarly, Löckenhoff and Carstensen [39] found that subjective well-being either increases or remains stable with age. Bartram [40] found a marginal increase in life satisfaction beyond middle age. One potential explanation for these contradictory findings is that certain areas of life satisfaction may compensate for declines in other areas, as overall life satisfaction is composed of multiple domains [1]. Regarding the link between education and life satisfaction, some studies find positive associations between the two variables [28,41], while others identify negative relationships [42]. Additionally, research highlights different mediators, such as the discrepancy between education and employment [43] and having educational aspirations that surpass the available opportunities [44], which can potentially mediate the negative impact of education on life satisfaction. Finally, the number of close friends is also positively associated with life satisfaction.

Cancer, which is a chronic disease that some parts of the body cells that grow uncontrollably and even spread to other parts of the body, tends to negatively affect life satisfaction. It is documented that people with cancer have a lower life satisfaction [45]. It is, thus, reasonable to suspect that cancer may influence the connection between personality and ratings on life satisfaction. These relationships are important to understand because life satisfaction is closely associated with morbidity and mortality outcomes. This present study seeks to establish the moderating role of cancer in the associations between the Big Five personality traits and life satisfaction.

## 2. Methods

### 2.1. Data

Data from Understanding Society: the UK Household Longitudinal Study (UKHLS), which has been collecting information on a yearly basis from the first sample of UK households beginning in 1991, was used in this research [46]. The University of Essex Ethical Committee approved all the research and data collection methods. All those who participated in the study gave their informed consent. In Wave 1, they all answered the question if they have been clinically diagnosed with cancer (collected between 2009 and 2010). Wave 2 and Wave 3 asked participants if they have been newly diagnosed with cancer. In Wave 3, participants answered questionnaires regarding personality, demographic, and life satisfaction (completed between 2011 and 2012). Participants who ever indicated that they have been diagnosed with cancer were regarded as people with cancer and people who never indicated that they have been diagnosed with cancer were considered as people without cancer. The information from those who could not complete all the questions was not included in the analysis. There were 1214 people with a diagnosis of cancer (38.55% males) had an average age of 59.70 (S.D. = 15.53) years and 13,319 people without a cancer diagnosis (38.13% males) had an average age of 59.97 (S.D. = 11.10) years.

### 2.2. Measures

#### 2.2.1. Cancer

Those who took part answered the question “Has a doctor or other health professional ever told you that you have any of these conditions? Cancer.” to indicate if they had cancer.

#### 2.2.2. Personality Traits

Participants’ personality traits were identified based on the 15-item version of the Big Five Inventory [47], using a Likert scale data collection instrument and utilizing a scale from one (“disagree strongly”) to five (“agree strongly”). This shorter form of the Big Five has sufficient validity, as shown by its internal consistency, test–retest correlations, and convergent and discriminant validity [48,49]. Where appropriate, the scores were reverse-coded. The list of questions asked of the participants is available here: https://www.understandingsociety.ac.uk/documentation/mainstage/dataset-documentation/term/personality-traits?search_api_views_fulltext= (accessed on 20 January 2023).

#### 2.2.3. Life Satisfaction

Those who participated were invited to answer the question “How dissatisfied or satisfied are you with… your life overall?”, with options consisting of a seven-point scale ranging from one (“not satisfied at all”) to seven (“completely satisfied”). Data from the single-item and multi-item measures including the Satisfaction with Life Scale (SWLS) were shown to be generally similar [28].

#### 2.2.4. Control Variables

Demographics included age, gender, monthly income, highest educational achievement, marital status, and number of close friends (Table 1).

### 2.3. Analysis

To consider if cancer influences the relationship between personality traits and life satisfaction, the hierarchical regression model [50] was used by entering variables into the model in four steps to estimate a person’s life satisfaction: (1) demographic variables including age, gender, monthly income, highest educational achievement, marital status, number of close friends; (2) personality traits; (3) cancer status; (4) personality traits by cancer status interactions. As a post hoc test, two multiple regression models were employed by using demographic variables including age, gender, monthly income, highest educational achievement, marital status, and personality traits, including Neuroticism, Agreeableness, Openness, Conscientiousness, and Extraversion, to predict life satisfaction in people who have and have not been diagnosed with cancer by a health professional. All analyses were performed using MATLAB 2018a.

## 3. Results

The presented descriptive statistics (Table 1) display various demographic characteristics in healthy controls and people with cancer. The sample exhibits a skew in terms of sex, with a higher representation of females in both groups. Additionally, a significant proportion of participants in both groups have an educational qualification below the college level and are married. Regarding the other variables, there are minimal differences observed between healthy controls and people with cancer. Both groups exhibit similar mean ages, monthly incomes, number of close friends, and personality traits scores, including Neuroticism, Agreeableness, Openness, Conscientiousness, and Extraversion.

Table 2 shows the results of the stepwise hierarchical regression, entering (1) demographics, (2) personality traits, (3) cancer status, (4) and personality traits by cancer status interactions as predictors to predict life satisfaction in sequence. The overall model fit generally increased by adding more predictors. As shown in step 4, the main finding in the current study is that clinically diagnosed cancer significantly moderates the association between Agreeableness and life satisfaction (*b* = 0.11, *p* < 0.05, 95% C.I. [0.02, 0.19]).

Figure 1 displays the relationship between Agreeableness and life satisfaction, separated by cancer status. The positive association between Agreeableness and life satisfaction is stronger in people with cancer compared to people without cancer.

Indeed, according to the results from the multiple regression (Table 3), Neuroticism is negatively related to life satisfaction in people with cancer (*b* = −0.22, *p* < 0.001, 95% C.I. [−0.28, −0.16]) and without cancer (*b* = −0.23, *p* < 0.001, 95% C.I. [−0.25, −0.21]). Agreeableness has a stronger positive relationship with life satisfaction in people with cancer (*b* = 0.17, *p* < 0.001, 95% C.I. [0.09, 0.26]) compared to people without cancer (*b* = 0.03, *p* < 0.05, 95% C.I. [0.003, 0.05]). Openness (*b* = 0.04, *p* < 0.001, 95% C.I. [0.02, 0.06]) and Conscientiousness (*b* = 0.11, *p* < 0.001, 95% C.I. [0.09, 0.14]) are positively associated with life satisfaction in people without cancer. Finally, Extraversion is positively related to life satisfaction in people with cancer (*b* = 0.08, *p* < 0.05, 95% C.I. [0.02, 0.15]) and without cancer (*b* = 0.06, *p* < 0.001, 95% C.I. [0.04, 0.08]).

## 4. Discussion

This research sought to understand how cancer can influence the association between personality traits and ratings of life satisfaction. This study indicated that cancer markedly influences the relationship between Agreeableness and life satisfaction, after accounting for demographic variables. Neuroticism was negatively associated with life satisfaction in people with and without clinically diagnosed cancer, whereas Agreeableness and Extraversion were positively associated with people with and without clinically diagnosed cancer. Openness and Conscientiousness were positively related to life satisfaction in people without cancer but were not significant predictors for people with cancer.

The main finding was that cancer moderates the association between Agreeableness and life satisfaction. with Agreeableness having a stronger relationship with life satisfaction in people with clinically diagnosed cancer. Individuals who are agreeable are polite and cooperative, which results in better social well-being [51], which may then lead to better life satisfaction, given that overall life satisfaction is made up of areas of life satisfaction. Moreover, agreeable people tend to engage in health-promoting behaviors [52,53,54]. Agreeableness may also contribute to positive life satisfaction through social support. In addition, this association was stronger in people with clinically diagnosed cancer, which may be consistent with the important role of Agreeableness in coping and adaptation following a chronic disease or disability [55]. Moreover, individuals who possess agreeable traits may have a higher propensity than individuals with disagreeable traits to adhere to instructions and advice after experiencing cancer, potentially leading to significant psychological advantages, which can explain why this relationship is stronger in people with cancer.

We found a negative relationship between Neuroticism and life satisfaction, which was aligned with previous studies (e.g., [1,2,14]). Individuals with high levels of Neuroticism are more emotionally unstable, have fewer positive emotions, and are usually unable to cope in challenging situations. In addition, Neuroticism is shown to predict activities related to health such as speed of walking, exercise levels, and a variety of physical and mental problems [56,57]. In addition, depression is also associated with elevated Neuroticism [58] and dementia [59]. The attitude and worldview of those with elevated Neuroticism are often more negative, and this influences how they interpret their life experiences, which may then lead to a lower rating of satisfaction with life.

We also found that Extraversion is a positive predictor of life satisfaction in people with and without cancer. Those who self-report elevated Extraversion are confident and social. Extroverts are more positive about their experiences and have an optimistic worldview, and this leads to higher ratings for life satisfaction. A characteristic of those who score high on Extraversion is that they are more physically active [60,61], sleep better [56], and are less likely to be depressed and anxious [58]. As a result, they are more likely to be satisfied with life.

Those who have an elevated level of Openness are more open to new experiences and have a broad range of interests and hobbies. They do not like routine but want novelty and have a positive worldview. They are always challenging themselves and want to find something interesting, which means that they are always seeking new stimulation [62]. Those who report elevated Openness are more adept at meeting their social and psychological needs, which influences their assessment of life satisfaction. Those who have the trait of Openness are more likely to possess and indeed enjoy a healthy lifestyle [56,57,60,63]. However, we failed to find such a relationship in people with cancer, which may be explained by the smaller sample size for people with cancer.

Conscientiousness is a trait characterized by being focused on tasks and maintaining order. Previous studies [1,2,14] show that this trait is positively associated with life satisfaction in both individuals with cancer and healthy individuals. Individuals with high Conscientiousness scores tend to actively seek out environments that support their goals and lead to greater accomplishments [18,64]. Their ability to achieve their objectives enhances their self-perception of competence and success, ultimately contributing to higher levels of life satisfaction [18,64]. However, it is important to note that the non-significant association between Conscientiousness and life satisfaction observed in individuals with cancer may be due to the smaller sample size of this specific population.

## 5. Limitations

This study has notable strengths, including the careful control of demographic variables and the inclusion of a group of healthy controls for comparison. However, it also has certain limitations that should be acknowledged. One limitation is that the data relied on self-reported information, which is subjective and carries a significant risk of bias. To enhance the validity of future research, it is recommended to incorporate more objective measures, such as clinical examinations or laboratory tests. Another concern is that the study adopts a cross-sectional design, which limits the ability to make causal inferences. To fully understand the causal relationships, a longitudinal approach would be beneficial, as it allows for the examination of changes over time. Additionally, this study fails to specify the specific types of cancer under investigation, which poses another limitation to its findings. Providing detailed information about the types of cancer would enhance the applicability and generalizability of the results. Finally, the effects were quite small, so interpretation must be accompanied with caution.

## 6. Implication

Health professionals should develop strategies and interventions by fostering personality traits including Agreeableness, Openness, Conscientiousness, and Extraversion, while reducing Neuroticism, which is proven to be feasible [65]. Doing so could improve people’s overall sense of well-being and their perception of their quality of life, which can help those diagnosed with cancer to have better outcomes.

## 7. Conclusions

To conclude, the goal of this study was to test if clinically diagnosed cancer moderates the associations between the Big Five personality traits and life satisfaction. This research found that cancer influences the association between Agreeableness and life satisfaction. Specifically, Neuroticism was negatively related to life satisfaction in people with and without cancer. Agreeableness had a stronger positive relationship with life satisfaction in people with cancer compared to people without cancer. Openness and Conscientiousness were positively associated with life satisfaction in people without cancer. Finally, Extraversion was positively related to life satisfaction in people with and without cancer.

## Figures and Tables

**Figure 1 healthcare-11-02359-f001:**
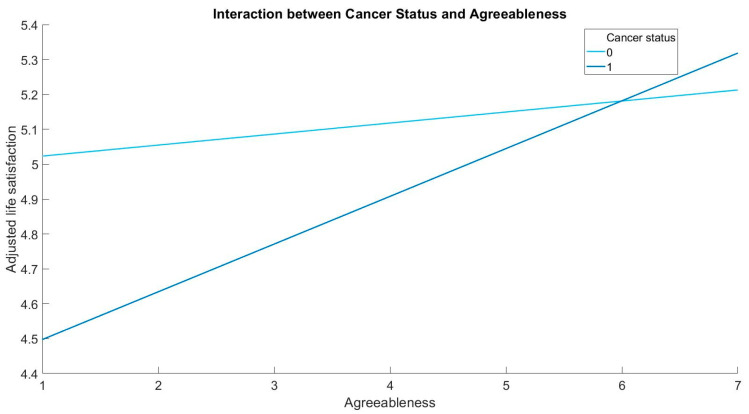
The moderating role of cancer status in the association between Agreeableness and life satisfaction. Figure 1 visually represents the significant interaction between Agreeableness and cancer status. If the two lines in the figure (representing those with and without cancer) are parallel, it would mean that Agreeableness has the same relationship with life satisfaction regardless of cancer status. However, as they are not parallel, it suggests that the relationship between Agreeableness and life satisfaction differs for individuals with and without cancer.

**Table 1 healthcare-11-02359-t001:** Descriptive statistics for people with and without cancer. The sample is skewed in terms of females, education below college, and marriage.

	Healthy Controls		People with Cancer	
	Mean	S.D.	Mean	S.D.
Age	59.97	11.10	59.70	15.53
Monthly income (£)	1461.89	1526.03	1419.79	1238.40
Number of close friends	5.29	5.66	5.64	5.84
Neuroticism	3.45	1.47	3.51	1.47
Agreeableness	5.73	1.03	5.73	1.07
Openness	4.45	1.36	4.47	1.37
Conscientiousness	5.61	1.11	5.50	1.14
Extraversion	4.55	1.36	4.62	1.39
	N	%	N	%
**Sex**				
Male	5078	38.13	468	38.55
Female	8241	61.87	746	61.45
**Highest educational qualification**				
Below college	9773	73.38	881	72.57
College	3546	26.62	333	27.43
**Legal marital status**				
Single	5051	37.92	527	43.41
Married	8268	62.08	687	56.59

**Table 2 healthcare-11-02359-t002:** The stepwise hierarchical regression results by entering these variables as predictors in sequence: (1) demographics, (2) personality traits, (3) cancer status, (4) and personality traits by cancer status interactions. All numbers are rounded up to two digits.

	*b*
** *Step 1* **	
Age	0.02 ***
Sex	0.12 ***
Monthly income	0.01 ***
Highest educational qualification	0.12 **
Legal marital status	0.40 ***
Number of close friends	0.02 ***
R^2 (adjusted R^2)	0.05 (0.05)
** *Step 2* **	
Age	0.02 ***
Sex	0.18 ***
Monthly income	0.01 ***
Highest educational qualification	0.09 **
Legal marital status	0.37 ***
Number of close friends	0.01 ***
Neuroticism	−0.23 ***
Agreeableness	0.04 ***
Openness	0.04 ***
Conscientiousness	0.11 ***
Extraversion	0.06 ***
R^2 (adjusted R^2)	0.13 (0.12)
** *Step 3* **	
Age	0.02 ***
Sex	0.18 ***
Monthly income	0.01 ***
Highest educational qualification	0.09 **
Legal marital status	0.37 ***
Number of close friends	0.01 ***
Neuroticism	−0.23 ***
Agreeableness	0.04 ***
Openness	0.04 ***
Conscientiousness	0.11 ***
Extraversion	0.06 ***
Cancer status	−0.02
R^2 (adjusted R^2)	0.13 (0.12)
** *Step 4* **	
Age	0.02 ***
Sex	0.18 ***
Monthly income	0.01 ***
Highest educational qualification	0.09 **
Legal marital status	0.37 ***
Number of close friends	0.01 ***
Neuroticism	−0.23 ***
Agreeableness	0.03 *
Openness	0.04 ***
Conscientiousness	0.11 ***
Extraversion	0.06 ***
Cancer status	−0.45
Neuroticism × cancer status	0.03
Agreeableness × cancer status	0.11 *
Openness × cancer status	−0.01
Conscientiousness × cancer status	−0.05
Extraversion × cancer status	0.02
R^2 (adjusted R^2)	0.13 (0.13)

**p* < 0.05, ***p* < 0.01, ****p* < 0.001.

**Table 3 healthcare-11-02359-t003:** The estimates (*b*) of multiple regression models for healthy controls and people who have been clinically diagnosed with cancer, by taking demographics and personality traits as the predictors and life satisfaction as the predicted variable. All numbers are rounded up to two digits.

	Healthy Controls	People with Cancer
Age	0.01 ***	0.01 *
Sex	0.20 ***	−0.08
Monthly income	0.01 ***	0.00
Highest educational qualification	0.09 **	0.14
Legal marital status	0.39 ***	0.19 *
Number of close friends	0.01 ***	0.00
Neuroticism	−0.23 ***	−0.22 ***
Agreeableness	0.03 ***	0.17 ***
Openness	0.04 ***	0.01
Conscientiousness	0.11 ***	0.07
Extraversion	0.06 ***	0.08 *
R^2	0.13	0.11

**p* < 0.05, ***p* < 0.01, ****p* < 0.001.

## Data Availability

Publicly available datasets were analyzed in this study. These data can be found here: https://beta.ukdataservice.ac.uk/datacatalogue/studies/study?id=6614 (accessed on 20 January 2023).

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
