# Peer review of "Understanding the Role of Cancer Diagnosis in the Associations between Personality and Life Satisfaction"

_healthcare, 2023, doi:10.3390/healthcare11162359_

Round 1
Reviewer 1 Report
I think that this is an interesting paper that is well presented. However, I think that the Methods and Results need some attention. First, I know you used hierarchical regression, but it is not clear the order in which the predictors were entered until you reach the discussion. In the discussion, you talk about adding the demographic variables first, but this needs to be explained earlier in the methods. Also, you presumably entered the personality traits in the regression before entering their interaction with cancer status, yet this is not mentioned in the analysis section of the methods. In the results, the change in R2 should be reported for each step of the regression. The results in Table 2 appear to refer to the multiple regressions that were run, but the results from the hierarchical regression should be reported. In fact, I'm not sure why you ran multiple regressions when the hierarchical regression provides all of the information you need to test the variables of interest. In the methods, you mention that information from those who could not complete all of the question was not included. How many were excluded and did you consider using multiple imputation to deal with missing data?
Small points that should be dealt with are monthly income (what was this denomiated in?), perhaps you should mention that the sample was skewed in terms of females, and the last sentence in the discussion doesn't make much sense. What does it mean to have social, psychological and behavioral outcomes? Also, in the discussion you should acknowledge that Openess was positively, but not significantly related to life satisfaction among people with cancer assuming that you even want to present the multiple regressions, because as I said earlier, this appears unecessary.
This appears fine.
Author Response
Dear Reviewer,
We thank you for reviewing this manuscript and providing constructive feedbacks. Here are our responses:
- We have reported the whole hierarchical regression supplied with tables and a figure. Now it should be more clear. In addition, 5% date points were removed due to missing. Multiple regression can tell about the direction and strength of the effect in patients and controls.
- We have added how the sample is skewed in terms of females, etc..
- We have changed the last sentence in the discussion.
- We have acknowledged nonsignificant associations between personality traits and life satisfaction.
Reviewer 2 Report
This manuscript aims to test the moderation effect of Cancer in the association between personality traits, using big five model and life satisfaction using data from UK Household Longitudinal Study.
I believe that, while the manuscript has several strengths, its main limitation regards to the statistical analysis. Authors aimed to test moderation; however, it seems that they carried out two multiple regressions in each group. Is important to note that this procedure does not prove moderation. Additionally, the statistical analysis section is hard to follow, and the presentation of the results require more information, such as standard errors and overall fit statistical tests.
Additionally, there are several sections that are hard to follow, as the grammar is a little awkward.
An extensive revision of the grammar is requiered.
Author Response
Dear Reviewer,
We thank you for your comments. Here are our responses:
- We have carefully clarified and reported the analysis.
- We have carefully reviewer and corrected the grammar.
Reviewer 3 Report
The study of association between Agreeableness and Life Satisfaction in People with cancer study is very important for understanding the factors of subjective well-being of cancer patients. This knowledge will make it possible to better implement psychological assistance to interested persons. The data obtained demonstrate a high level of connection between life satisfaction and agreeableness in cancer patients, which may act as a kind of compensator for the lack of positive experiences.
At the same time, a strong (tenfold) imbalance of samples causes some tension. Perhaps the results of the study would be more convincing if the samples were aligned by the number of respondents. It would also be desirable to enter the proportion of variance described by one or another variable (ΔR2).
In our opinion, it is necessary to clearly state in the work a conclusion that allows us to understand the difference between the two samples, which follows from the results obtained, and try to explain it psychologically.
I wish success to the authors in publishing this material.
Author Response
Dear Reviewer,
We thank you for reviewing this manuscript and providing feedbacks. Here are our responses:
- We have added the step by step results of the hierarchical regression. We felt the sample size is fine as there is no requirement for equal sample size in hierarchical regression. We also noticed the potential impact of the sample size in the discussion.
- We have added a separate conclusion section.
Reviewer 4 Report
After reviewing the document, I suggest that you work on the following points to consider its publication:
Review the wording of the abstract. The abstract should be reworded (life satisfaction is very repetitive)
The element of originality is not detected in the proposed abstract
2.3. analysis
It is important to clarify why they are using this regression technique. For what type of endogenous and exogenous variables is it suggested?
3. Results
The proposed interval values do not always include the estimated parameter value, which is clearly wrong. Review again the estimated parameters and the confidence intervals constructed from them. They are supposed to be symmetric about the estimated value.
It would be a good idea for each of the variables included in the model to be theoretically justified for its inclusion, that is, to indicate its support in the literature. It also seems to me that the resulting R2 statistic has a very low value, does the software also show the adjusted R2?
I did not find the mention about the software used for the estimation of the models
Discussion
In the last paragraph of the discussion, why is it concluded only in relation to the Agreeableness variable when at least 2 other components, according to what they report, also turned out to be significant?
To make it clearer, divide the information reported in the discussion into sections:
Conclusion
Practical implications of the study
Limitations and lines of future research
References
Update the references since there are very few that are less than 5 years old.
After addressing these comments, more elements will be available to make a better decision regarding this work.
Minor editing of English language required
Author Response
Dear Reviewer,
We thank you for reviewing this manuscript. Here are our response:
- We have reworded the abstract.
- We have clarified the analysis.
- We have carefully checked the confidence interval.
- We have justified why demographics are included as control variables.
- We have mentioned the results of other traits and reorganized the structure of the discussion.
- We have updated some of the references. However, we kept some old references because they are key papers in life satisfaction research field.
Round 2
Reviewer 1 Report
It was very difficult to follow what the authors had done. Simply providing their response without knowing what I had recommended made things difficult. Providing the manuscript in track changes to see what changes have been made is essential. However, I can see that they have provided the hierarchical regression and talked about the missing data as well as justified the multiple regressions. They should talk about the skewed nature of the sample (males and females) in the limitation.s Also, further proofreading is necessary as the standard of English is not that good.
Further proofreading of the manuscript is required as the standard of English is not that good.
Author Response
Dear Reviewer,
Thank you for you advice. We have discussed the descriptive statistics of the sample and corrected the English.
Reviewer 2 Report
This is the second version of the manuscript. Even when statistical analysis is more clear it does not respond to the moderation role that is described in the introduction. Therefore this results are inconsistent to what is intended to perform.
Author Response
Thank you. Interaction is a valid proof of moderation as we cited.
Reviewer 4 Report
The authors must prepare a document responding point by point to the observations made by the reviewers so that their follow-up is easier.
The authors are supporting their findings in a hierarchical regression model; however I believe that they have not adequately justified the use of the technique. For what type of variables is it recommended? Why is this the right technique for your study?
The adjusted coefficient of determination is still very low, what implications does this have for your conclusions according to the literature on the regression technique?
They should write some points of their document in a better way (for example, 2.2.4. Control variables).
Author Response
Dear Reviewer,
Thank you for your advice. Here are our responses:
(1) Hierarchical regression model can test moderation as we cited.
(2) We have recognized the limitation.
(3) We have proofread the paper.